

# Synthetic dietary inulin, Fuji FF, delays development of diet-induced obesity by improving gut microbiota profiles and increasing short-chain fatty acid production

Miki Igarashi[1,*], Miku Morimoto[1,*], Asuka Suto[1], Akiho Nakatani[1], Tetsuhiko Hayakawa[1], Kenjirou Hara[2] and Ikuo Kimura[1]

[1] Department of Applied Biological Science, Graduate School of Agriculture, Tokyo University of Agriculture and Technology, Fuchu, Tokyo, Japan
[2] Fuji Nihon Seito Corporation, Shizuoka, Japan
[*] These authors contributed equally to this work.

## ABSTRACT

**Background**. Dietary fiber, including inulin, promotes health via fermentation products, such as short-chain fatty acids (SCFAs), produced from the fiber by gut microbiota. SCFAs exert positive physiological effects on energy metabolism, gut immunity, and the nervous system. Most of the commercial inulin is extracted from plant sources such as chicory roots, but it can also be enzymatically synthesized from sucrose using inulin producing enzymes. Studies conducted on rodents fed with a cafeteria diet have suggested that while increasing plasma propionic acid, synthetic inulin modulates glucose and lipid metabolism in the same manner as natural inulin. Therefore, this study aimed to determine the effects of a synthetic inulin, Fuji FF, on energy metabolism, fecal SCFA production, and microbiota profiles in mice fed with a high-fat/high-sucrose diet.

**Methods**. Three-week-old male C57BL/6J mice were fed a high-fat/high-sucrose diet containing cellulose or Fuji FF for 12 weeks, and the effects on energy metabolism, SCFA production, and microbiota profiles were evaluated.

**Results**. Body weight gain was inhibited by Fuji FF supplementation in high-fat/high-sucrose diet-fed C57BL/6J mice by reducing white adipose tissue weight while increasing energy expenditure, compared with the mice supplemented with cellulose. Fuji FF also elevated levels of acetic, propionic and butyric acids in mouse feces and increased plasma propionic acid levels in mice. Moreover, 16S rRNA gene amplicon sequencing of fecal samples revealed an elevated abundance of Bacteroidetes and a reduced abundance of Firmicutes at the phylum level in mice supplemented with Fuji FF compared to those supplemented with cellulose. Fuji FF also resulted in abundance of the family Bacteroidales S24-7 and reduction of Desulfovibrionaceae in the feces.

**Conclusion**. Long term consumption of Fuji FF improved the gut environment in mice by altering the composition of the microbiota and increasing SCFA production, which might be associated with its anti-obesity effects.

Corresponding authors
Miki Igarashi,
mikiigarashi@go.tuat.ac.jp
Ikuo Kimura, ikimura@cc.tuat.ac.jp

## INTRODUCTION

The prevalence of overweight and obese individuals is increasing globally along with that of metabolic diseases such as type II diabetes mellitus (T2DM), hepatic steatosis, and several types of cancer (*Chooi, Ding & Magkos, 2019*; *Collaboration, 2017*). Although the etiology of obesity is thought to involve a complex interplay between genetic and environmental factors, diet is considered the most important contributor to the increased worldwide incidence of obese and overweight individuals (*Chan & Woo, 2010*; *Chooi, Ding & Magkos, 2019*). Dietary fat intake has become elevated due to the increased consumption of processed and fast foods, as the intake of dietary fiber that can provide health benefits such as lowered risk of diseases including obesity and diabetes has decreased (*Delzenne et al., 2020*; *Hadrevi, Sogaard & Christensen, 2017*). These benefits of dietary fiber are conferred by disrupting the absorption of nutrients in the gut, while useful fibers are fermented by microbiota. Dietary fiber can also change the composition of the microbiota to reduce risk of diseases including obesity. Therefore, a dietary fiber that promotes the growth of beneficial microorganisms in the intestines, is referred to as a prebiotic, and its consumption is recommended to maintain host health (*Gibson et al., 2017*).

Various plants naturally produce inulin, which is a dietary fiber (*Mensink et al., 2015*) comprising straight chains of repetitive fructosyl moieties that are linked by $\beta(2 \rightarrow 1)$ bonds with a $(1 \leftrightarrow 2)$ D-glucosyl moiety at the terminus (*Mensink et al., 2015*). The length of these fructose chains varies, and the degree of polymerization (DP) is usually between 2 and 60 (*Mensink et al., 2015*). Inulin is a polysaccharide applied in the food, pharmaceutical and chemical industries because of its unique variety of physicochemical properties (*Mensink et al., 2015*). Accumulating evidence has emphasized the importance of dietary fiber to the prevention and treatment of obesity (*Delzenne et al., 2020*). Inulin exerts prebiotic anti-obesity effects because it is fermented to SCFA, especially butyric acid, by gut microbiota (*Hoving et al., 2018*). These effects seem to depend not only on the dietary dosage, but also on the DP of inulin (*Van Loo, 2004*). Differences in the DP also impact inulin fermentation and the intestinal region where it occurs, and this changes the composition of metabolites and the microbiome in intestinal segments (*Le Blay et al., 2003*). Current research seeks to understand the differences in the prebiotic functions of inulin with different DP or lengths (*Le Blay et al., 2003*).

Commercial inulin powders are extracted from plants such as chicory, and typically contain 6%–10% monosaccharides as impurities from the native source (*Coussement, 1999*; *Mensink et al., 2015*). However, inulin synthesized from glucose and fructose contains few monosaccharides (*Wada et al., 2005*). The synthetic inulin (average DP = 16–18) prevented body weight gain in rodents fed with a high-fat diet for 12 weeks and reduced lipid accumulation in the rat liver while reducing plasma triacylglycerol levels. Furthermore, blood glucose levels were reduced more in rodents fed with a high-fat and high-sucrose diet for three weeks by a synthetic inulin (average DP = 16–18) than by another inulin with an average DP between 6 and 8 and natural inulin with a DP of 23, suggesting that the length of the synthetic inulin affects its properties and physiological functions like that of plant-sourced inulin (*Sugatani et al., 2008*; *Wada et al., 2005*). However, information about

whether dietary synthetic inulin affects energy metabolism has not been reported; only the results of one study have suggested that synthetic inulin could cause the proliferation of gut bacteria in vitro, and might affect intestinal bacteria composition (*Wada et al., 2005*). Therefore, we investigated whether the synthetic inulin, Fuji FF, could prevent the development of diet-induced obesity by modifying the composition of the gut microbiota and SCFA production in vivo.

## MATERIALS & METHODS

### Study design and experiments

All animal experiments proceeded in accordance with the guidelines of the Committee on the Ethics of Animal Experiments of the Tokyo University of Agriculture and Technology, and were approved by the Animal Research Ethics Subcommittee (Permit number: 28–87). Three-week-old male C57BL/6J mice (Japan SLC Inc., Hamamatsu City, Shizuoka, Japan) were acclimated for one week to animal facility conditions (temperature, 22 °C–24 °C; humidity, 30%–60%), under a controlled 12 h light/12 h dark cycle with free access to water and the CE-2 standard rodent diet (CLEA Japan Inc., Tokyo, Japan). The mice were randomly assigned at the age of four weeks, to receive diets containing either cellulose or Fuji FF for 12 weeks ($n = 18$ per group) and were housed two per cage. The diets were formulated based on the D08112601M high-fat/high-sucrose diet (Research Diets Inc., New Brunswick, NJ, USA) comprising 45% kcal fat and 30% kcal sucrose, and supplemented with either 10% (w/w) of 38-$\mu$m (400 mesh) cellulose powder (FUJIFILM Wako Pure Chemical Corporation, Osaka, Japan), or Fuji FF (Fuji Nihon Seito Corporation, Tokyo, Japan). The mice were weighed and food intake was measured weekly for 12 weeks, and respiratory quotient, energy expenditure and locomotive activity were assessed between weeks 9 and 11 as described below. The mice were fasted for 5 h at week 12 while fresh fecal droppings were collected and blood glucose was measured using a portable glucometer with compatible glucose test strips (OneTouch® Ultra®, LifeScan Inc., Milpitas, CA, USA). Thereafter, the mice were anesthetized using Somnopentyl® (Kyoritsu Co. Ltd., Tokyo, Japan). Blood was collected from the inferior vena cava using heparinized tubes, and plasma was separated by centrifugation (7,000 g, 5 min, 4 °C). Livers, ceca, kidneys, testes, and epididymal, perirenal and subcutaneous white adipose tissues were harvested and weighed. Tissues, feces and plasma samples were stored at −80 °C.

### Measurement of energy metabolism and locomotion

Individually housed mice in a metabolic chamber were acclimatized for at least for 12 h between weeks 9 and 11. We then measured $VO_2$ and $VCO_2$ using an MK-5000RQ device and locomotive activity using CompACT AMS ver.3 software (both from Muromachi Kikai Co. Ltd., Tokyo, Japan). The system was controlled under a strict 12-h light/dark cycle under atmospheric conditions of 22 °C and 30%–60% humidity. Measurements proceeded for at least 48 h including the acclimation period ($\geq 12$ h) with free access to food and water. Respiratory quotients and energy expenditure were calculated based on $VO_2$ and $CO_2$ values determined using MMS-4 version 6.2 (MK-5000RQ operating software). Continuous 24 h data about the respiratory quotient, energy expenditure and locomotive

activity were extracted. We measured energy metabolism and locomotion in 12 and 10 mice given cellulose and Fuji FF, respectively, due to the limitations of our system and time constraints imposed by the experimental schedule.

## Quantitation of short-chain fatty acids

Fecal samples ($\geq$50 mg) were suspended in water (10 w/v) then separated by centrifugation (8,000 g, 5 min, 4 °C). Fecal supernatants (350 $\mu$L) or plasma (80$\mu$L) sample was incubated with 5-sulfosalicylic acid (2% vol) at 4 °C for 15 min and separated by centrifugation (15,000 g, 15 min, 4 °C). Fecal (300 $\mu$L) or plasma (50 $\mu$L) supernatant was mixed with 2-ethylbutyric acid (30 pmol; internal standard), hydrochloric acid (5% vol), and diethyl ether (200% vol). The samples were vigorously mixed and separated by centrifugation (3,000 g, 5 min). The organic layer was transferred to GC vials and applied for SCFA analysis using a gas chromatography-mass spectrometer, a GCMS-QP2010 Ultra (Shimadzu Corporation, Kyoto, Japan). The GC was fitted with a VF-WAXms 30-m capillary column with an internal diameter of 0.25 mm and a film thickness of 0.36 $\mu$m (Agilent Technologies Inc., Santa Clara, CA, USA). Helium was the carrier gas at a flow rate of 1 mL/min. Samples (2 $\mu$L) were injected in splitless mode. The initial oven temperature was maintained at 50 °C for 1 min; ramped to 145 °C, 220 °C and 240 °C in increments of 15 °C/min, 8 °C/min and 15 °C/min, respectively, then held at 240 °C for 1 min. The temperatures of the EI ion source and injector were 200 °C and 250 °C, respectively. The electron energy was 70 eV. Full-scan mass spectra were recorded in the 40–90 m/z range. Amounts of SCFA were quantified by integration of the extracted ion chromatographic peaks for the following ion species: m/z 60 for acetic acid at 9.7 min, m/z 74 for propionic acid at 11.3 min, m/z 60 for butyric acid at 12.5 min, and m/z 88 for 2-ethylbutyric acid at 14.2 min. The absolute levels of SCFA were quantified using calibration curves of individual SCFA and 2-ethylbutyric acid.

## Analysis of fecal microbiota by 16S rRNA gene amplicon sequencing

Fecal DNA was extracted from frozen samples using FastDNA® SPIN Kits for Feces (MP Biomedicals LLC., Irvine, CA, USA) according to the manufacturer's instructions. The V4 region of the 16S rRNA gene was amplified with dual-indexed 515F/806R primers. The library concentration was measured using a KAPA Library Quantification kit designed for Illumina platforms (Kapa Biosystems Inc., Wilmington, MA, USA). The amplicon size was determined using an Bioanalyzer High Sensitivity DNA kits (Agilent Technologies, Santa Clara, CA, USA). The amplicons were sequenced using an Illumina MiSeq system with a MiSeq Reagent 222 kit V3 (Illumina, San Diego, CA, USA). Libraries with concentrations of 1.0 pM were prepared using the ''Preparing Libraries for Sequencing on the MiSeq'' (part 15039740, Rev. D) protocol. The16S microbial sequencing data were analyzed using Quantitative Insights into Microbial Ecology (QIIME) software version 1.9.1 (http://www.quiime.org). Paired reads were stitched with paired-end read merger (PEAR) and further filtered based on the quality of identification using Phred quality scores (QN19). Chimeric reads were excluded using USEARCH61, then filtered reads were demultiplexed within QIIME, and samples with <5,000 reads were excluded from

further analysis. Representative raw sequences were further aligned using Python Nearest Alignment Space Termination with the SILVA core-set alignment template (Silva version 128) to obtain bacterial operational taxonomic units (OTU). The raw OTU data (relative abundance) was filtered by domain (bacteria only), and eliminated based on the average of the abundance of each bacterial species (<0.0001 for phylum and class levels, <0.00001 for order and family levels, <0.000001 for genus level). Unclassified and other data were discarded for further analysis at each level, which were then re-normalized. Principal component analysis (PCA) plots were generated using the function prcomp in the R package to identify clustering within each level Compositional similarity was compared using the permutational multivariate analysis of variance (PERMANOVA), the R Vegan package. We measured microbiome composition in seven data samples per group, but one data sample for the cellurose group was removed due to insufficient reads. The raw data have been deposited into the DNA Data Bank of Japan (DDBJ) database Accession no. DRA008499.

### Plasma lipids

Plasma total cholesterol and triglyceride levels were measured using LabAssay$^{TM}$ Cholesterol and LabAssay$^{TM}$ Triglyceride kits (FUJIFILM Wako Pure Chemical Co.) as described by the manufacturer.

### Statistical analysis

Data are expressed as means $\pm$ SEM or as means $\pm$ SD and were statistically analyzed using GraphPad Prism version 7.04 software (GraphPad Software Inc., La Jolla, CA, USA). Normality and the homogeneity of variance were tested using Shapiro–Wilk and F-tests, respectively. Mann–Whitney $U$ tests were applied to paired comparisons if the normality assumption was violated, and unpaired Aspin-Welch tests were applied if the heterogeneity of variance was significant. Unpaired $t$-tests were applied if statistical assumptions were met. Statistical significance in multiple comparisons was evaluated using unpaired Student $t$-tests or two-way analyses of variance (two-way ANOVA) followed by Bonferroni multiple comparison tests, as appropriate. Significance between groups was established at adjusted $p < 0.05$.

## RESULTS

### Fuji FF reduced HF/HS diet-induced increases in body and adipose tissue weight

Mouse body weight and food intake were monitored weekly during 12 weeks of feeding with diets supplemented with either cellulose or Fuji FF. Figure 1A shows that significantly less weight was gained by the Fuji FF, than the cellulose group after six weeks ($p = 0.012$ and $P < 0.001$ at 6 and 7–12 weeks, respectively). Figures 1B and 1C respectively shows that weekly (g/cage/week) and total (g/cage/12 weeks) food intake did not significantly differ between the cellulose and Fuji FF groups (mean average ($\pm$SD) energy intake, 32.0 $\pm$ 6.9 and 27.5 $\pm$ 4.9 kcal/cage/day, respectively). At the end of the feeding period, the epididymal, perirenal, and subcutaneous adipose tissues weighed significantly less in the

mice supplemented with Fuji FF than with cellulose ($p < 0.001$ for all), whereas the cecum weight increased ($p < 0.001$; Fig. 1D). Blood glucose levels did not significantly differ between the two groups after a 5-h fast (Fig. 1E). Supplementation with Fuji FF reduced levels of total cholesterol ($p < 0.001$), but not of triacylglycerol ($p = 0.6071$), in mouse plasma (Figs. 1F and 1G). These results indicated that Fuji FF inhibited the increase in body weight induced by feeding with the high-fat/ high-sucrose diet, presumably due to reducing fat accumulation.

## Fuji FF affected energy metabolic and locomotive activity

We compared energy expenditure and locomotive activity in mice supplemented with Fuji FF or cellulose, because Fuji FF reduced the body and adipose tissue weight without altering food intake (Fig. 1). The respiratory quotient (RQ) was lower during the dark period in mice supplemented with Fuji FF than with cellulose (Figs. 2A and 2B; $p = 0.0448$, $p = 0.0344$, $p = 0.0128$ and $p = 0.003$ at 23:00 h, at 00:00 h, at 01:00 h, and the dark periods, respectively), suggesting that mice supplemented with Fuji FF metabolized more fat than carbohydrate or protein. On the other hand, the mice supplemented with Fuji FF expended significantly more energy during both the dark and light periods (Figs. 2C and 2D; $p = 0.006$ at 08:00 h, $p < 0.001$ at 21:00 and 22:00 h, $p = 0.0015$ at 23:00 h, $p = 0.0263$ at 00:00 h, $p = 0.019$ at 07:00, and $p = 0.003$ and $p = 0.014$ during the dark and light period, respectively). The mice supplemented with Fuji FF also spent more time engaging in locomotive activity than those supplemented with cellulose, indicating that the increased energy usage was due to physical activity (Figs. 2E and 2F; $p < 0.001$ at 21:00 and 22:00, $p = 0.012$ at 23:00 and $P < 0.001$ during the dark period). These results indicated that Fuji FF prevented fat accumulation in mice fed with the high-fat/high-sucrose diet by increasing fat oxidation and energy expenditure.

## Fuji FF increased fecal and plasma SCFAs

We measured fecal and plasma SCFAs in the mice supplemented with cellulose or Fuji FF for 12 weeks because SCFAs generated from inulin are thought to be major influences on host energy metabolism (Fig. 3). Levels of acetic, propionic and butyric acids (Figs. 3A, 3B and 3C, respectively) were significantly elevated in the feces of the mice supplemented with Fuji FF compared with cellulose ($p < 0.001$, $p < 0.001$ and $p = 0.028$, respectively). In addition, plasma propionic acid was significantly increased (Fig. 3E) in the mice supplemented with Fuji FF ($p = 0.039$), whereas acetic (Fig. 3D) and butyric (Fig. 3F) acids were not. Thus, Fuji FF increased the intestinal production of SCFA via fermentation, which then led to increased plasma levels of propionic acid, perhaps because most SCFAs produced in the intestine were metabolized in the liver.

## Fuji FF changed gut microbiota composition

Fuji FF supplementation affected bacterial populations at the phylum level in the mouse gut. The relative abundance of Bacteroidetes was elevated in the Fuji FF, compared with the cellulose group ($P = 0.0082$), whereas the relative abundance of Firmicutes and Proteobacteria was decreased ($P = 0.0069$ and $P = 0.0095$, respectively; Fig. 4A). Fuji FF did not affect the relative abundance of any other phyla that we measured. Further investigation

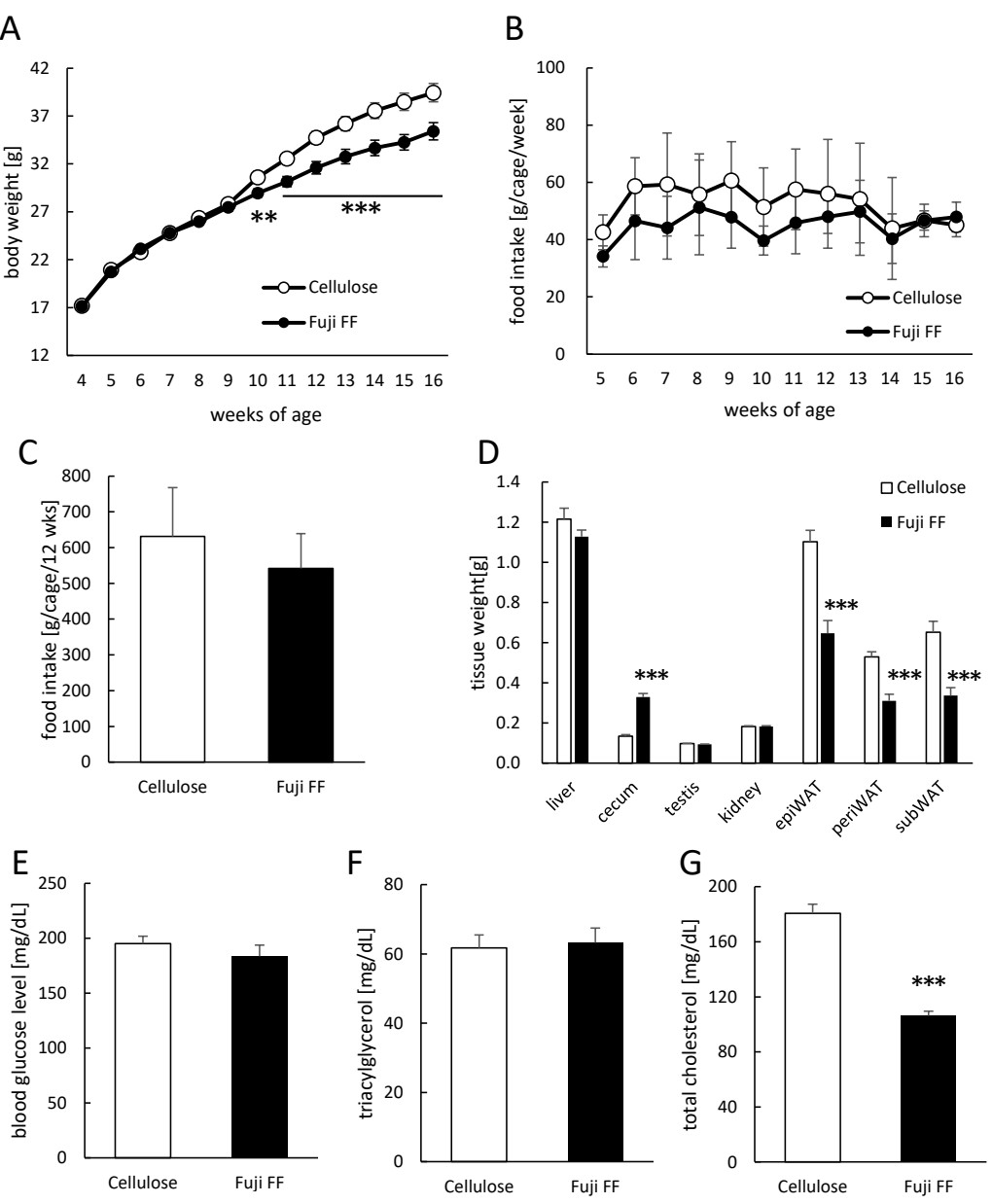

**Figure 1** **Effects of Fuji FF on body weight gain, food intake, tissue weight, blood glucose, and plasma lipids in mice.** (A) Body weight gain. (B) Weekly food intake (g/cage/week; $n = 9$). (C) Total food intake (g/cage/12 week; $n = 9$). (D) Weight of liver, cecum, kidney, testis, and white epididymal (epiWAT), perirenal (periWAT) and subcutaneous (subWAT) adipose tissues in mice fed with high-fat/high sucrose diet supplemented with either cellulose or Fuji FF for 12 weeks. (E) Blood glucose, (F) plasma triacylglycerol and (G) plasma total cholesterol values measured in mice after fasting for 5 h. Data are expressed as means ± SEM (A, D, E, F and G) or as means ± SD (B and C). Significance is established at adjusted $p < 0.05$ (*$p < 0.05$, **$p < 0.01$, and ***$p < 0.001$). Cellulose group: $n = 18$ for A, D and E–G; Fuji FF group: $n = 18$ for A, D and E, $n = 16$ for F, $n = 17$ for G. Each cage contained two mice.

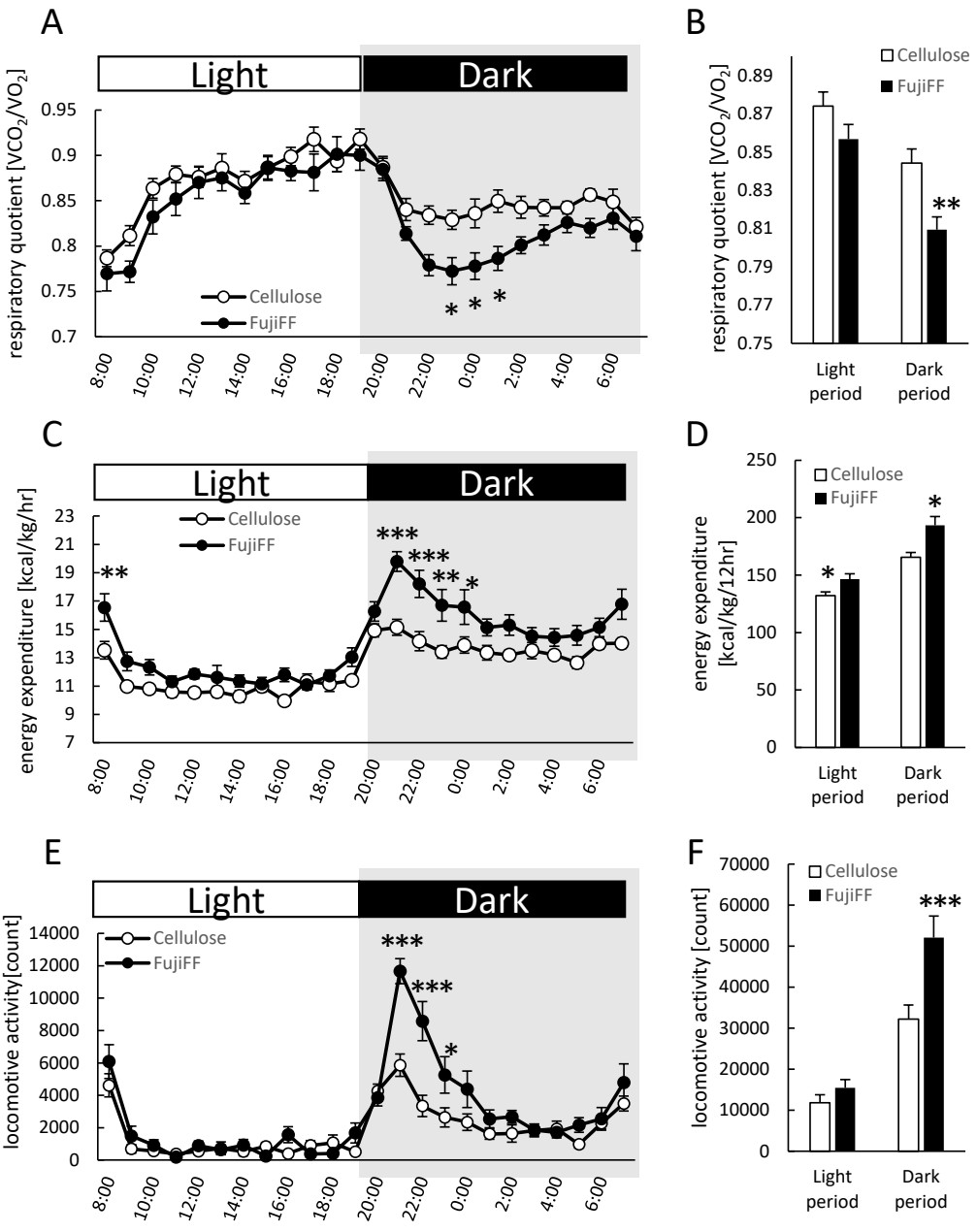

**Figure 2** **Effects of Fuji FF on respiratory quotient, energy expenditure, and locomotive activity in mice.** Respiratory quotient (A and B), energy expenditure (C and D), and locomotive activity (E and F) measured in mice fed supplemented with cellulose or Fuji FF for 9 weeks. Hourly changes in respiratory quotient (A), energy expenditure (C), and locomotive activity (D) over 24 h. Dark and light cycles are indicated as gray and white backgrounds, respectively. Average of respiratory quotient (B), energy expenditure (D), and locomotive activity (E) during dark and light cycles. Data are expressed as means ± SEM ($n$ = 10–12). Significance is established at adjusted $p < 0.05$ (*$p < 0.05$, **$p < 0.01$, and ***$p < 0.001$).

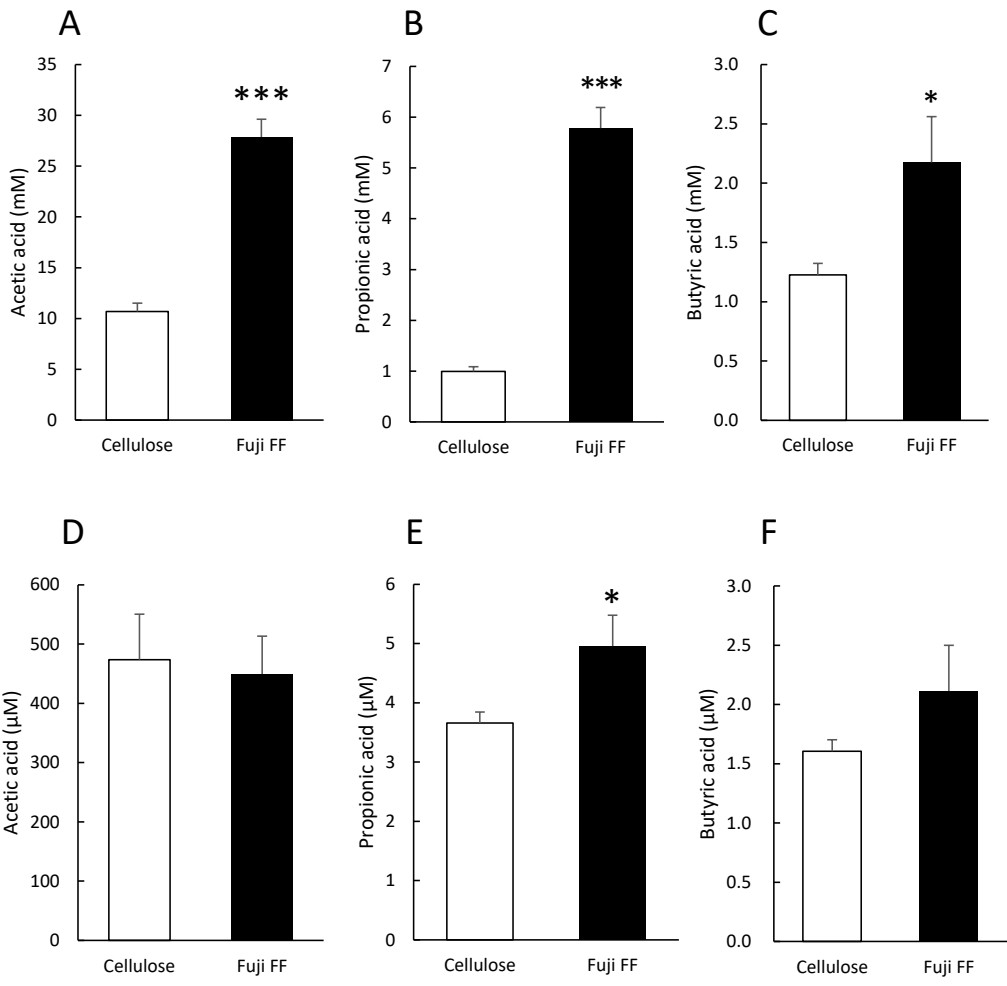

**Figure 3** **Fuji FF increases short-chain fatty acid contents in feces and plasma.** Amounts of (A and D) acetic, (B and E) propionic and butyric (C and F) acids among SCFA extracted from fecal (A–C) samples collected during a 5-h fast after 12 weeks of cellulose or Fuji FF supplementation and plasma (D–F) samples collected from the inferior vena cava determined using GC-MS. Data are expressed as means ± SEM ($n = 18$). Significance is established at adjusted $p < 0.05$ (*$p < 0.05$ and ***$p < 0.001$).

using principal component analyses (PCA) indicated significant segregation between the groups at the phylum (Fig. 4B), class (Fig. 4C), order (Fig. 4D), family (Fig. 4E) and genus (Fig. 4F) levels, $p = 0.008$, $p = 0.003$, $p = 0.002$, $p = 0.002$ and, $p = 0.001$, respectively. In addition, linear discriminant analysis (LDA) and the effect size (LEfSe) algorithm identified taxa with the greatest differences in abundance at each level between the cellulose and the Fuji FF groups. A bar chart (Fig. 5A) and cladogram (Fig. 5B) show the results of linear discriminant analysis (LDA) and the effect size (LEfSe) algorithm, respectively, of fecal microbiota at family levels in mice supplemented with Fuji FF or cellulose. At the family level, the cellulose-associated microbiota was characterized by Streptococcaceae and Peptostreptococcaceae belonging to the phylum Firmicutes, and Desulfovibrionaceae belonging to the phylum Proteobacteria, whereas the Fuji FF-associated microbiota

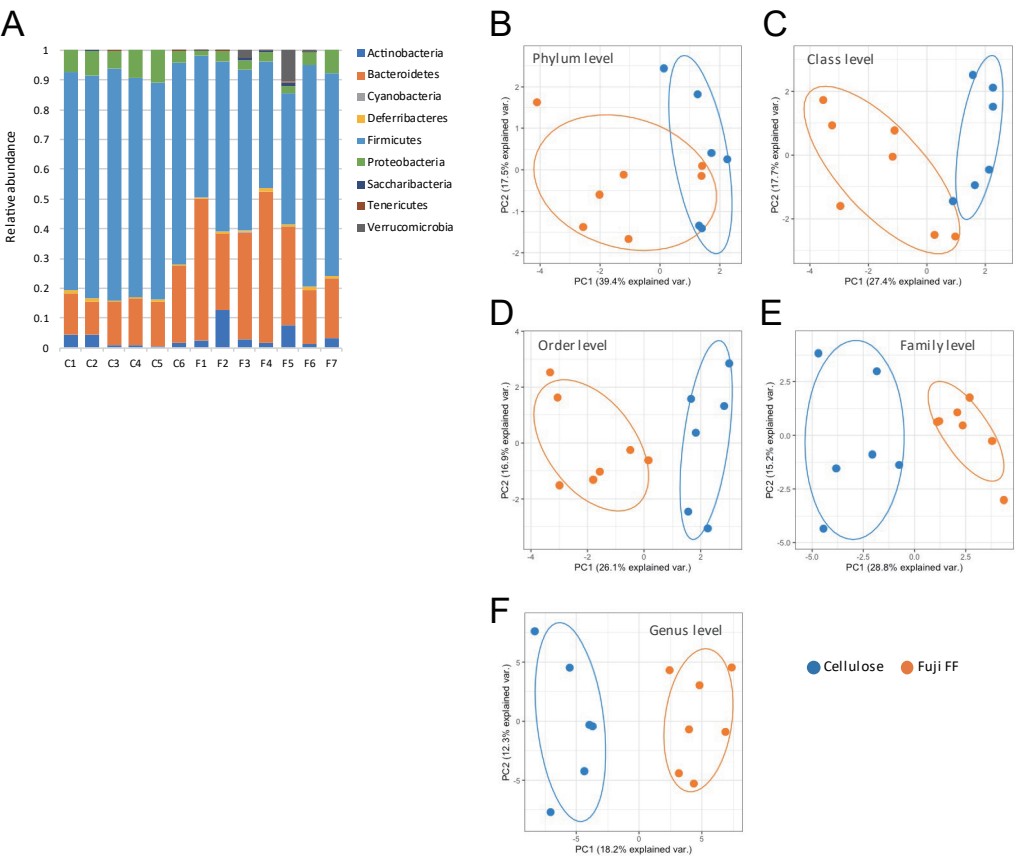

**Figure 4  Changes in fecal microbiota of mice supplemented with Fuji FF or cellulose.** (A) Relative abundance of major taxonomic groups at phylum level, and (B–E) principle component analysis (PCA) of taxonomic groups at phylum (B), class (C), order (D), family (E) and genus (F) levels of fecal microbiota in mice supplemented with Fuji FF or cellulose. $n = 6$ for cellulose and $n = 7$ Fuji FF groups.

was characterized by Bacteroidales S24-7 and Peptococcaceae belonging to the phyla Bacteroidetes and Firmicutes, respectively. Further, at the family level, Bacteroidales S24-7 (Fig. 5C, $p = 0.0040$) and Peptococcaceae (Fig. 5E, $P = 0.0043$) were significantly elevated, whereas Desulfovibrionaceae (Fig. 5D, $p = 0.0011$), Peptostreptococcaceae (Fig. 5E, $p = 0.021$) and Streptococcaceae (Fig. 5E, $p < 0.001$) were significantly reduced in the fecal microbiota of the mice supplemented with Fuji FF. Bifidobacteriaceae, which ferments fibers to SCFAs, did not significantly differ between the two groups. These results collectively indicated that dietary Fuji FF significantly modified the gut microbiome profile.

## DISCUSSION

This study investigated whether the synthetic inulin, Fuji FF, exerts anti-obesity effects like other types of inulin, by improving the intestinal environment in mice fed with a high-fat/high-sucrose diet. The results indicated that Fuji FF attenuated body weight gain and fat accumulation, while increasing energy expenditure. Fuji FF also modified the

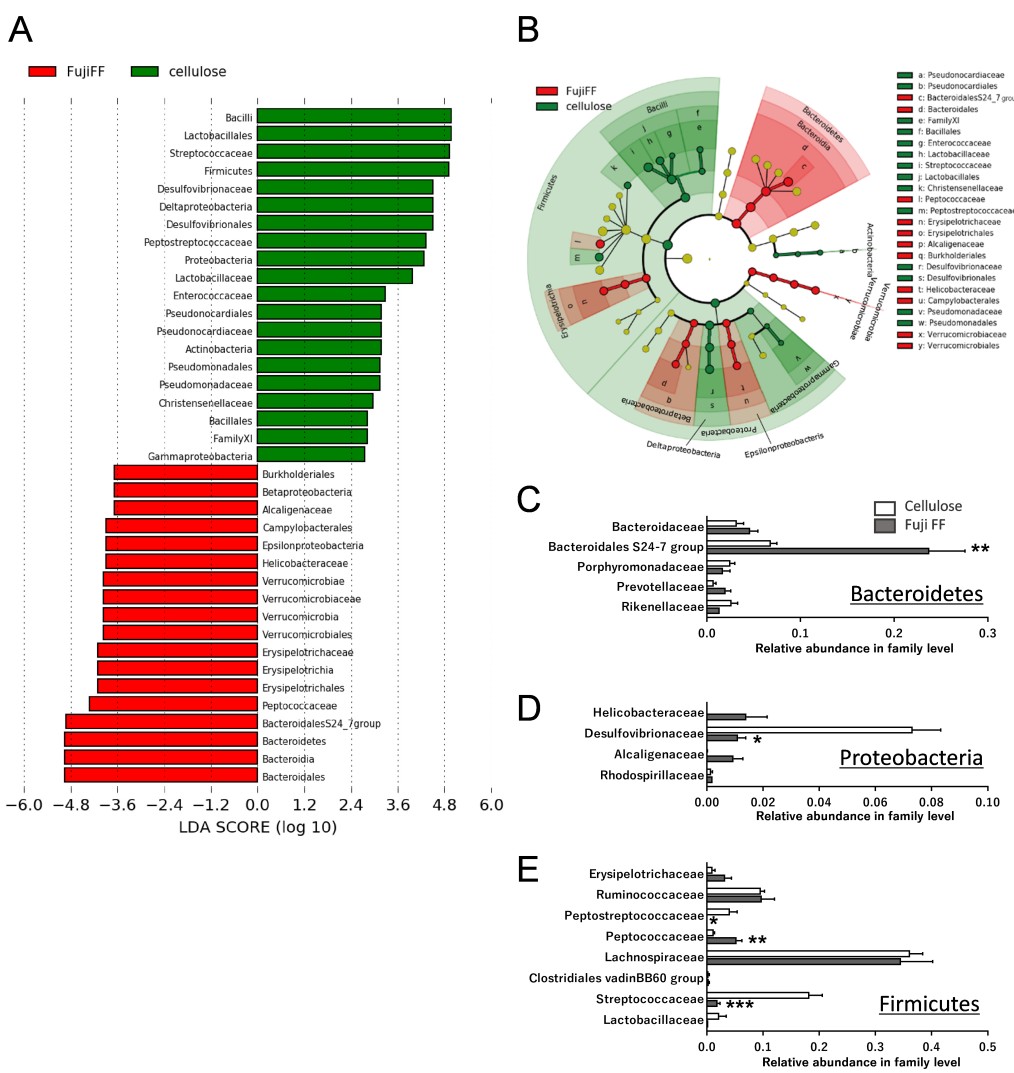

**Figure 5** **Changes in fecal microbiota at family levels of mice supplemented with Fuji FF or cellulose.**
(A) Bar chart and (B) cladogram of results of linear discriminant analysis (LDA) and effect size (LEfSe) algorithm of fecal microbiota at family levels in mice supplemented with Fuji FF or cellulose. Relative abundance of major microbiota at family level in Bacteroidetes (C), Proteobacteria (D), and Firmicutes (E). Data are expressed as means ± SEM ($n = 6$ for cellulose and $n = 7$ for Fuji FF groups). Significance is established at adjusted $p < 0.05$ (*$p < 0.05$, and **$p < 0.01$).

microbiota composition, then increased the amounts of SCFA in feces and plasma. These findings showed that Fuji FF can improve the gut environment in vivo and prevent the development of diet-induced obesity diet in mice.

Like other inulin and fiber formulations, Fuji FF inhibited gains in body weight caused by the high-fat/high-sucrose diet in mice. Most dietary fibers, including inulin, are not digested in the small intestine, but are metabolized by the microbiota in the large intestine (*Macfarlane & Macfarlane, 2012*). The major SCFA products of microbial fermentation of fibers in the gut are acetic, propionic and butyric acids (*Cummings et al., 1987*; *Matt et al.,*

*2018*). Through specific receptors, SCFAs play important roles in the maintenance of health, energy metabolism and the prevention of certain diseases (*Alexander et al., 2019*; *Kimura et al., 2020*). One study has suggested that inulin consumption increases the butyrate content in the cecum and feces of animals (*Hoving et al., 2018*). Another study also suggested that synthetic inulin supplementation increases levels of plasma propionic acid in rats fed with a high-fat and high-sucrose diet (*Sugatani et al., 2008*). The SCFAs produced from fiber are thought to be responsible for preventing diet-induced obesity and improving energy homeostasis (*Weitkunat et al., 2017*). In addition, SCFA supplementation inhibits weight gain in mice with diet-induced obesity (*Lin et al., 2012*). Notably, Fuji FF elevated levels of acetic, propionic and butyric acids in the feces of mice fed a high-fat/high-sucrose diet, but increased only propionic acid in plasma, which supports previous findings (*Sugatani et al., 2008*) (Fig. 3). These results indicated that Fuji FF can be assimilated to provide SCFAs by microbiota, and that it could prevent diet-induced obesity via the functions of SCFAs.

Fuji FF obviously reduced fat accumulation in mice by increasing fat oxidation and energy expenditure as reflected by a lower RQ and more energy expenditure. Notably, inulin and oligofructose do not affect fat excretion in the small intestine of patients with an ileostomy (*Ellegard, Andersson & Bosaeus, 1997*), although the effect might depend on the dose of fibers. Therefore, increased levels of SCFA induced by Fuji FF might be major modulators of enhanced energy metabolism because SCFAs are endogenous ligands of G protein-coupled receptor (GPR) 41 and GPR43 (*Kimura et al., 2011*), which are involved in many physiological processes, including energy metabolism and neurological functions (*Kimura et al., 2014*). The sympathetic ganglia of mice and humans notably express abundant GPR41; thus, energy expenditure and body temperature might be influenced by SCFAs via GPR41 that regulates activation of the sympathetic nervous system (*Kimura et al., 2011*). In addition, GPR43 is expressed in adipose tissues, where it controls fat accumulation. The phenotype of GPR43-deficient mice is obese, whereas that of mice overexpressing GPR43 in adipose tissues is lean/normal (*Kimura et al., 2013*). Further investigation is needed to clarify how these receptors are involved in the health benefits of Fuji FF.

The gut microbiota is an important factor that can regulate host energy metabolism and behavior (*Tremaroli & Backhed, 2012*). Additionally, the gut microbiota converts dietary fiber to SCFA, which increases the abundance of health-promoting bacteria and reduces the abundance of potentially pathogenic bacteria (*Zentek et al., 2003*). The ratio of Firmicutes to Bacteroides is higher in animals with genetically or diet-induced obesity, although this remains controversial (*Ley et al., 2005*; *Moschen, Wieser & Tilg, 2012*). We previously found that various dietary supplements such as fiber-like materials reduce this ratio in experimental animal models of obesity (*Nakatani et al., 2018*; *Watanabe et al., 2018*). Our findings of the 16S rRNA gene amplicon sequences revealed that the ratio of Firmicutes to Bacteroides was reduced more by Fuji FF than by cellulose. Our results at the family level showed an elevated relative abundance of Bacteroidales S24-7 groups, which are thought to be anti-diabetic, in the phylum Bacteroides (*Krych et al., 2015*). In addition, Fuji FF reduced the relative abundance of Desulfovibrionaceae, which is positively associated

with obesity and/or T2DM and produces an endotoxin (*Hildebrandt et al., 2009*; *Zhang et al., 2010*). Furthermore, increased Peptococcaceae and Bacteroidales S24-7 at the family level by Fuji FF, are associated with SCFA production (*Org et al., 2017*; *Smith et al., 2019*). Therefore, Fuji FF supplementation can positively change gut microbiota composition by increasing levels of SCFA, resulting in protection against diet-induced obesity.

## CONCLUSIONS

This study investigated whether the synthetic inulin, Fuji FF, could prevent the development of diet-induced obesity by modifying the composition of the gut microbiota and SCFA production in vivo.

Our findings showed that Fuji FF has a powerful capacity to prevent diet-induced obesity by modifying the intestinal environment through fermentation to SCFAs in mice. The potential role of SCFAs in the anti-obesity action of Fuji FF inulin should be further explored by evaluating their effects on GPR41 and GPR43 receptors.

## ACKNOWLEDGEMENTS

We thank Fuji Nihon Seito Corporation for supplying Fuji FF.

### Funding

This research was supported by Fuji Nihon Seito Corporation. The funders had no role in study design, data collection and analysis, decision to publish, or preparation of the manuscript.

### Grant Disclosures

The following grant information was disclosed by the authors:
Fuji Nihon Seito Corporation.

### Competing Interests

Kenjirou Hara is an employee of Fuji Nihon Seito Corporation, which is a manufacturer of Fuji FF. All other authors declare that they have no competing interests.

### Author Contributions

- Miki Igarashi conceived and designed the experiments, performed the experiments, analyzed the data, prepared figures and/or tables, authored or reviewed drafts of the paper, and approved the final draft.
- Miku Morimoto and Asuka Suto performed the experiments, analyzed the data, prepared figures and/or tables, and approved the final draft.
- Akiho Nakatani and Tetsuhiko Hayakawa performed the experiments, prepared figures and/or tables, and approved the final draft.
- Kenjirou Hara analyzed the data, authored or reviewed drafts of the paper, and approved the final draft.

- Ikuo Kimura conceived and designed the experiments, analyzed the data, authored or reviewed drafts of the paper, and approved the final draft.

### Animal Ethics

The following information was supplied relating to ethical approvals (i.e., approving body and any reference numbers):

The Committee on the Ethics of Animal Experiments of the Tokyo University of Agriculture and Technology provided approval for this research (permit number: 28–87).

### DNA Deposition

The following information was supplied regarding the deposition of DNA sequences:

The raw data are available in the DNA Data Bank of Japan (DDBJ): DRA008499.

### Data Availability

The raw data is available in the Supplemental Files.

### Supplemental Information

Supplemental information for this article can be found online at http://dx.doi.org/10.7717/peerj.8893#supplemental-information.

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
