# Peer review of "Synthetic dietary inulin, Fuji FF, delays development of diet-induced obesity by improving gut microbiota profiles and increasing short-chain fatty acid production"

_PeerJ, doi:10.7717/peerj.8893_

## Round 0.1 · original submission · Major Revisions

Please, try to address all the reviewers comments carefully.

Reviewer 1 ·

Basic reporting

The objective of this study was to evaluate the potential beneficial effects of the administration of the synthetic inulin, Fuji FF on energy metabolism and in the modulation of the gut microbiota by employing an in vivo rodent model of diet induced obesity. The manuscript follows the structure of the journal although English should be revised because some parts though the text are not clear. In addition, there are some aspects that should be improved and revised to improve the quality of the article.
Abstract: Line 32-43 are not clearly explained. Previous data indicate that synthetic inulin has beneficial effects in glucose and lipid metabolism and modifies gut microbiota but I cannot understand why the authors include in the sentence the comparison with maltodextrin.
The literature employed in introduction and discussion should be improved and extended because the study of complex carbohydrates non digested by the host but fermented by the gut microbiota it has been extensively studied in the last years. The term prebiotic should be also described through the text, including the most recent definition for this term (G.R. Gibson et al., Expert consensus document: The International Scientific Association for Probiotics and Prebiotics (ISAPP) consensus statement on the definition and scope of prebiotics, Nat. Rev. Gastroenterol. Hepatol., vol. 14, no. 8, pp. 491–502, 2017).
In addition, the importance of the employ of dietary fiber such as inulin in the prevention and treatment of obesity should be critically documented. There are recent review articles on this topic that they should be cited (As example: Delzenne et al., Nutritional interest of dietary fiber and prebiotics in obesity: Lessons from the MyNewGut consortium, Clin Nutr. 2019 Mar 9. pii: S0261-5614(19)30115-3 .2019). The authors only select some articles but I would like to know if there are no more recent articles studying synthetic inulin and if it is not the case they should clearly indicate it. The importance of SCFA in energy metabolism and potential mechanism of action should be better explained because it is currently an active research area.
The figures are appropriate and generally well described but some mistakes should be corrected and some improvements made:
Figure 4A: Please write correctly abundande
Figure 4 D,E, F: The families should we written in another colour, for example black because the grey colour is not easy to read

Experimental design

The article covers the scope of the journal and the research questions are clearly stated. The tests performed answered only some of the objectives of the article.
Methods are not well described and detailed in some cases:
Although the objective of the study is to evaluate differences of the administration of cellulose and Fuji FF inulin in a murine model of diet-induced obesity I would have included another third group that received only the high fat/ high sucrose diet to compare. Please also indicate why they have supplemented the high fat/ high sucrose diet with 10% of cellulose or Fuji FF. This dose has been previously tested?
Please indicate the method employed for SCFA measurement in blood. The authors have only described the method employed for SCFA quantification in feces.
Since the amount of food ingested by the two groups was not different it would be helpful to know the daily energy intakes of the different groups.
I understand to check differences in the weight of the adipose tissue from the different areas but please better explained why they have also taken liver, caecum, kidney and testis.
Concerning the gut microbiota analysis, Have the authors not seen statically significant changes in the Phylum Actinobacteria or at lower taxonomic levels in the same phylum comparing both bacterial groups? If you check Figure 4A the relative abundance of Actinobacteria seems to change and be higher in Fuji FF group. Moreover, in the heatmap Bifidobacteriaceae is enriched in Fuji FF group and this group is normally modified by certain prebiotic supplements such as fructooligosaccharides, including in this group inulin.
SCFA profile: The authors observed differences in the concentration of the main SCFAs in faeces and blood but please indicate if the fecal SCFA results are for example concordant or not with the gut microbiota profiles obtained in Fuji FF group.
The statistical analysis should be better explained. The authors indicate that they have employed parametric tests but it is not clear if they have checked normality of the samples.

Validity of the findings

The experiments performed do not cover some of the data indicated in the results and conclusion sections.
Moreover, it is not clear why the authors have decided to test the Fuji FF synthetic inulin and why they have chosen the cellulose substrate to compare. It may be necessary a brief more accurate justification of the importance of the study: What is new?. I would like to know if Fuji FF inulin has been previously tested in another in vitro or in vivo models to evaluate the novelty of the article.
To understand the objective of the article it is important to clearly explain what are prebiotics in the introduction section and the importance of the study of prebiotics to treat obesity.
Some results are not clear and more experimentation should be performed. For example, in line 227 the authors indicate that there is no observable effect on insulin resistance but they have not measured insulin or performed HOMA-IR index or additional tests such as insulin tolerance test to affirm that. They have only measured fasted glucose levels in both groups. This sentence is too speculative and I would remove it.

Please indicate that the employ of the ratio Firmicutes/ Bacteroidetes in obesity is controversial. Different data from animal and human studies do not corroborate the results obtained from Ley et al. 2005.

The authors conclude that the employ of Fuji FF inulin is useful to prevent diet-induced obesity by modifying the composition and the metabolic activity of gut microbiota but for the experimentation performed it is too speculative to indicate that the responsible factors are SCFAs. I would modify the sentence “that SCFAs are likely the responsible factor for the benefits of Fuji FF…” and indicate that the potential role of SCFAs in the anti-obesity action of Fuji FF inulin should be further explored by evaluating the effects of SCFA in GPR41 and GPR43 receptors.

Additional comments

Other comments and some typos and grammatical errors:
Please standarize the text, sometimes sentences are written in present, another time in past to facilitate the reading.
Line 32 please change insulin producing enzyme for inulin producing enzyme….
Line 89-92: Please rewrite the sentence, it is not clear. Please indicate the specific results obtained in plasma after the administration of synthetic inulin.
Line 87: Please change insulin for inulin…
Line 98: Please change ani-obesity for anti-obesity.
Line 99: I would change elements of the gut environment for gut microbiota. The authors have not measured markers of inflammation, gut permeability or examine histologically the gut, They have focused in the gut microbiota profile.
Line 256-258: Please rephrase the sentence. It is not clear.
Line 278: I would say that dietary Fuji FF significantly modifiers or reshapes the gut microbiota instead of Fuji FF significantly improves…
Line 286: I would change improved for modulated…

Reviewer 2 ·

Basic reporting

The manuscript was written in clear and unambiguous english, and supplied with professional article structure, figures and tables. But some improvements are needed.
The introduction needs more details. I suggested supply more detailed information about Fuji FF, such as the average DP and so on, and summarized reported researches about Fuji FF.

Experimental design

The detailed procedure of plasma SCFAs measurement was not stated.

Validity of the findings

1. Raw data about Figure 4 should be supplied.
2. PCA of taxonomic groups at all levels should be analyzed, not only at family level.

Additional comments

1. Levels of acetic acid, propionic acid and butyric acid were elevated in the feces of the mice fed a Fuji FF diet, but only propionic acid was increased in the plasma. please explained why.
2. Fuji FF diet reduced the body weight and white adipose tissue, how Fuji FF affected the plasma lipids?
3. At the genus level, if there are some different fecal microbiota were identified?
4. Fuji FF diet increased the cecum weight, please explained why.
5. The authors should stated what's novelty the study added.

Reviewer 3 ·

Basic reporting

The manuscript by Igarashi and colleagues evaluate the effect of the synthetic inulin Fuji FF in delaying the metabolic and physiological effects associated with a high-sucrose/high fat western diet. Authors monitor weight gain and energy intake over 12 weeks and energy expenditure at week 9 of treatment. They also studied the gut microbiota profiles associated with the different dietary interventions, as well as some bacterial-derived metabolites. The major finding of this study is that animals supplemented with synthetic inulin (Fuji FF) get less weight over the time, reduce the amount of white adipose tissue and improve metabolic parameters compared with cellulose supplementation. These changes seem to be associated with specific gut microbiota compositional profiles that produce more SCFAs.
Authors provide a sufficient background of the current understanding and an extensive description of the methodology used; the later being highly appreciated by this reviewer. The manuscript is well written and the data clearly presented. However, there are a number of issues that should be clarified by the authors before recommending this paper for publication.

Experimental design

My major criticism to this work is that strictly speaking, authors do not show that Fuji FF inhibited the development of obesity (line 44), reduce white adipose tissue and improve energy expenditure (line 44-45). This is because authors did not induce metabolic syndrome in these animals (e.g. by feeding all the animals with only high-sucrose/high fat diet) first and after the establishment of metabolic syndrome, they did include the supplementations in the diet. This approach would be more similar to a real-world scenario in which a dietary intervention is set up to treat a patient with metabolic syndrome. Given this limitation, at the best, authors can conclude that the supplementation with Fuji FF delays the development of obesity. This limitation should be discused and the claims made reviewed accordingly.
Animals in this study exhibited a similar weight gain until week 6 of treatment. From week 7 onwards, animals supplemented with Fuji FF significantly gained less weight. This seems not to be associated with the caloric intake. Energy consumption in the Fuji FF animals trended to be reduced (although not statistically significant) compared to the cellulose arm. However, authors provided the overall energy intake. I would like to see if Fuji FF suplemmented animals ate less from week 6 onwards. Therefore authors should present energy intake over the course of the experiment. Accordingly with the methods section (lines 120-121) authors measured food intake weekly and each cage contained 2 animals. Thus, authors should have enough data to longitudinally provide a mean, SD and appropriate statistical comparison, of energy intake over the course of the experiment.

Additional comments:
1- Why did authors use really young (3-weeks old) animals?
2- Why unclassified OTUs were eliminated from the final OTU table? Was the percentage of these unclassified OTUs similar between interventions?
3- Which method did the authors use to normalize the OTU table?
4- In figure 4 A, include the bar chart for all of the samples, instead of using an average composition plot.
5- Figure 2B. Which distance matrix was used for the calculation of the PCA? please include in the figure legend. Authors should also perform a PERMANOVA analysis to complement figure 2B.
6- Figure 2C does not provide any useful information. I recommend the authors performing a LEfSe analysis to identify discriminant feature between interventions. Authors can then plot the abundance of these discriminant OTUs between treatments. This will provide the reader a more clear picture about which OTUs are associated with each dietary supplementation.

Validity of the findings

No comment

Additional comments

I have found some typos that should be amended:
Line 98: ani-obese should be anti-obese.

---

## Round 0.2 · accepted · Accept

The reviewers find that the manuscript has been improved and meets the standards of the journal.

Reviewer 2 ·

Basic reporting

The author answered the question reasonably.

Experimental design

The author answered the question reasonably.

Validity of the findings

The author answered the question reasonably.

Additional comments

Generally, the author answered the question reasonably. It would be better to discuss the possible mechanism that how Fuji FF affect the plasma lipids as well as why only propionic acid was increased in the plasma of the mice fed a Fuji FF diet.

Reviewer 3 ·

Basic reporting

The manuscript has been improved and in my eyes meet the standards of the journal.

Experimental design

Methods are appropriate and clearly explained. They have substantially been improved after resubmission.

Validity of the findings

All the claims made by the authors are supported by the data and figures presented.

Additional comments

The authors have addressed all my comments and I am happy to recommend this paper for publication.